# Formation of Aligned α-Si3N4 Microfibers by Plasma Nitridation of Si (110) Substrate Coated with $SiO_2$

**Chang-Hua Yu [1],\*, Kun-An Chiu [1], Thi-Hien Do [1], Li Chang [1],\* and Wei-Chun Chen [2]**

[1] Department of Materials Science and Engineering, National Yang Ming Chiao Tung University, Hsinchu 30010, Taiwan; j73628.mse95g@nctu.edu.tw (K.-A.C.); dohienvl@gmail.com (T.-H.D.)

[2] National Applied Research Laboratories, Taiwan Instrument Research Institute, Hsinchu 30010, Taiwan; weichun@tiri.narl.org.tw

\* Correspondence: z3aptx4869@yahoo.com.tw (C.-H.Y.); lichang@cc.nctu.edu.tw (L.C.); Tel.: +886-3-5712121 (ext. 55373) (L.C.); Fax: +886-3-5724727 (L.C.)

**Abstract:** Plasma nitridation of an amorphous $SiO_2$ layer on Si (110) substrate can form well-aligned α-$Si_3N_4$ crystallites in fibrous morphology. Nitriding is performed at a temperature in the range of 800–1000 °C by using microwave plasma with a gas mixture of $N_2$ and $H_2$. Raman spectroscopy shows the characteristics of an α-$Si_3N_4$ phase without other crystalline nitrides. As shown by scanning electron microscopy, the formed α-$Si_3N_4$ microfibers on the Si substrate can be in a dense and straight array nearly along with Si <$1\bar{1}0$>, and can have a length over 2 mm with a diameter in the range of 5–10 μm. Structural characterization of scanning transmission electron microscopy in cross section view reveals that the elongated α-$Si_3N_4$ crystallites are formed on the surface of the nitrided $SiO_2$/Si (110) substrate without any interlayers between $Si_3N_4$ and Si, and the longitudinal direction of α-$Si_3N_4$ appears mainly along <$11\bar{2}0$>, which is approximately parallel to Si <$1\bar{1}0$>.

**Keywords:** $Si_3N_4$; plasma; nitridation; fiber; electron microscopy

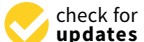



## 1. Introduction

Silicon nitride ($Si_3N_4$) has been used in many structural applications for a long time due to its excellent properties, such as high hardness and high-temperature stability, resistance to thermal shock, and wear resistance. In addition, $Si_3N_4$ has a wide band gap ($E_g$ = 4.7 eV) which can be applied for semiconductor devices [1,2]. $Si_3N_4$ has many different crystal structures, among which α-$Si_3N_4$, β-$Si_3N_4$, and γ-$Si_3N_4$ are common phases. Both α-$Si_3N_4$ and β-$Si_3N_4$ can be formed under ambient conditions at high temperatures, while γ-$Si_3N_4$ exists under high temperature and high pressure [3–5]. In addition, crystalline $Si_3N_4$ synthesis, generally in polycrystalline microstructure, has been widely conducted by nitridation of Si and $SiO_2$ powders at high temperature. Plasma nitriding has been proven to be effective for material synthesis [6–8]. Plasma nitridation of $SiO_2$ and Si for the formation of $Si_3N_4$ have been studied in previous reports by using thermal and radio frequency plasma processes with gases containing nitrogen [9–11] in which hydrogen may be necessary for the oxide reduction, and nitrogen reaction with Si formed nitride. It is well known that microwave plasma can easily achieve high plasma density which has been applied in various applications, including etching and film deposition. In our recent studies, we have shown that microwave plasma can be effectively applied for nitridation of *c*-plane and m-sapphires to form epitaxial AlN. In addition, epitaxial TiN can obtain plasma nitridation of rutile $TiO_2$ [12–14].

Various forms of crystalline $Si_3N_4$ have been synthesized, including whiskers and nanowires which may have potential applications in nanocomposites and electronic devices. $Si_3N_4$ nanowires have been synthesized by various methods, including chemical vapor deposition and nitridation [15–18]. Recently, we have used microwave plasma for nitridation of a $SiO_2$/Si (111) substrate to form {$10\bar{1}0$}-oriented crystalline α-$Si_3N_4$ and

β-Si₃N₄ which have elongated wire morphologies in lengths of several hundred microme­ters and are mainly aligned with three different Si <110> directions [19]. In this study, the formation of crystalline Si₃N₄ on an Si (110) substrate which was coated with SiO₂ was investigated. Furthermore, we try to understand the role of Si orientation on the alignment of Si₃N₄. The results showed that crystalline α-Si₃N₄ microfibers in aligned arrays with a length more than 2 mm are mainly formed along a *unique* direction of Si <1$\bar{1}$0> on Si (110).

## 2. Materials and Methods

In this work, a 50 nm thick SiO₂ film grown on a two inch Si (110) wafer was used as substrate. Growth of the SiO₂ film was performed at 300 °C by plasma-enhanced chemical vapor deposition using a gas mixture of SiH₄, N₂O, and Ar. After the deposition of SiO₂, the substrate was cut to pieces about 1 × 1 cm². Nitridation of the SiO₂/Si (110) substrate with a gas mixture of N₂ and H₂ was performed in a microwave (2.45 GHz) plasma system, which was commonly employed for chemical vapor deposition of diamond [20,21]. The flow rate of N₂ and H₂ was 285 and 15 sccm, respectively. The microwave power was set at 700 W and a pressure of $5.3 \times 10^3$ Pa. The nitridation was carried out for 10 and 20 min. The substrate temperature during plasma nitriding was estimated to be in the range of 800–1000 °C as measured by optical pyrometer (wavelength of 0.96 μm).

Raman spectroscopy was carried out in a Horiba Jobin Yvon LabRAM HR 800 spec­trometer with 50 mW Ar laser (λ = 533 nm). Scanning electron microscopy (SEM, JEOL JSM-6500F, Tokyo, Japan). was performed to examine surface morphologies of the SiO₂/Si (110) substrates. Cross-sectional scanning transmission electron microscopy (ARM200F, JEOL, Tokyo, Japan) with high-resolution transmission electron microscopy (HRTEM) was performed for crystallographic evaluation. Cross-sectional STEM specimen preparation was performed in a dual-beam focused ion beam (FIB, TESCAN LYRA3) system. Before FIB cutting, a 50 nm thin layer of amorphous carbon (a-C) and a 240 nm layer of platinum (Pt) were deposited on the nitride sample surface with an electron beam, followed by deposition of a 250 nm layer of a-C and a 300 nm layer of Pt with ion beam as protective coating.

## 3. Results

Figure 1a shows a typical SEM micrograph of the nitrided SiO₂/Si (110) substrate after 10 min nitriding. The surface morphology shows that Si₃N₄ crystallites exhibit a fiber-like characteristic with an average diameter of about 1–2 μm and a length of >0.1 mm. Those Si₃N₄ microfibers are predominately aligned in parallel with each other and approximately along the specific direction of Si [1$\bar{1}$0]. For 20 min nitridation, the length of the microfibers can be as long as 2 mm and the diameter can increase to 5–10 μm, as shown in Figure 1b,c. A few areas on the nitrided substrate are covered with highly dense microfibers. The microfibers shown in Figure 1c exhibit a prism-like morphology, and most of the microfibers are aligned nearly with their longitudinal directions Si [1$\bar{1}$0] in addition to a few deviated ones. The faceted morphology suggests that the microfibers are of the crystalline phase. Furthermore, X-ray energy dispersive spectroscopy shows that those microfibers consist of Si and N, as shown in Figure 1d, with the absence of O, implying that SiO₂ may not exist anymore after nitriding.

Raman spectra from different areas on the SEM images shown in Figure 1 were acquired in micromode. A typical Raman spectrum acquired from the microfibers is shown in Figure 2 from 150 to 450 cm⁻¹. The Raman peaks observed at 259 and 363 cm⁻¹ are identified to belong to α-Si₃N₄ [22]. The Raman peak at 300 cm⁻¹ is the characteristic signal of Si in second order [23,24]. As no β-Si₃N₄ characteristic peaks are observed, it is evident that all the microfibers are of the α-Si₃N₄ phase. On the bare area without coverage of the Si₃N₄ crystallites as shown in Figure 1b, neither Si₃N₄ nor SiO₂ Raman signals can be observed, implying that SiO₂ after nitridation might transform to Si₃N₄, or be etched away by the plasma containing hydrogen. Many of the glass vibrations are

known to occur between 400–700 cm$^{-1}$ (bridging Si-O-Si vibrations) which are not seen in the spectrum [24].

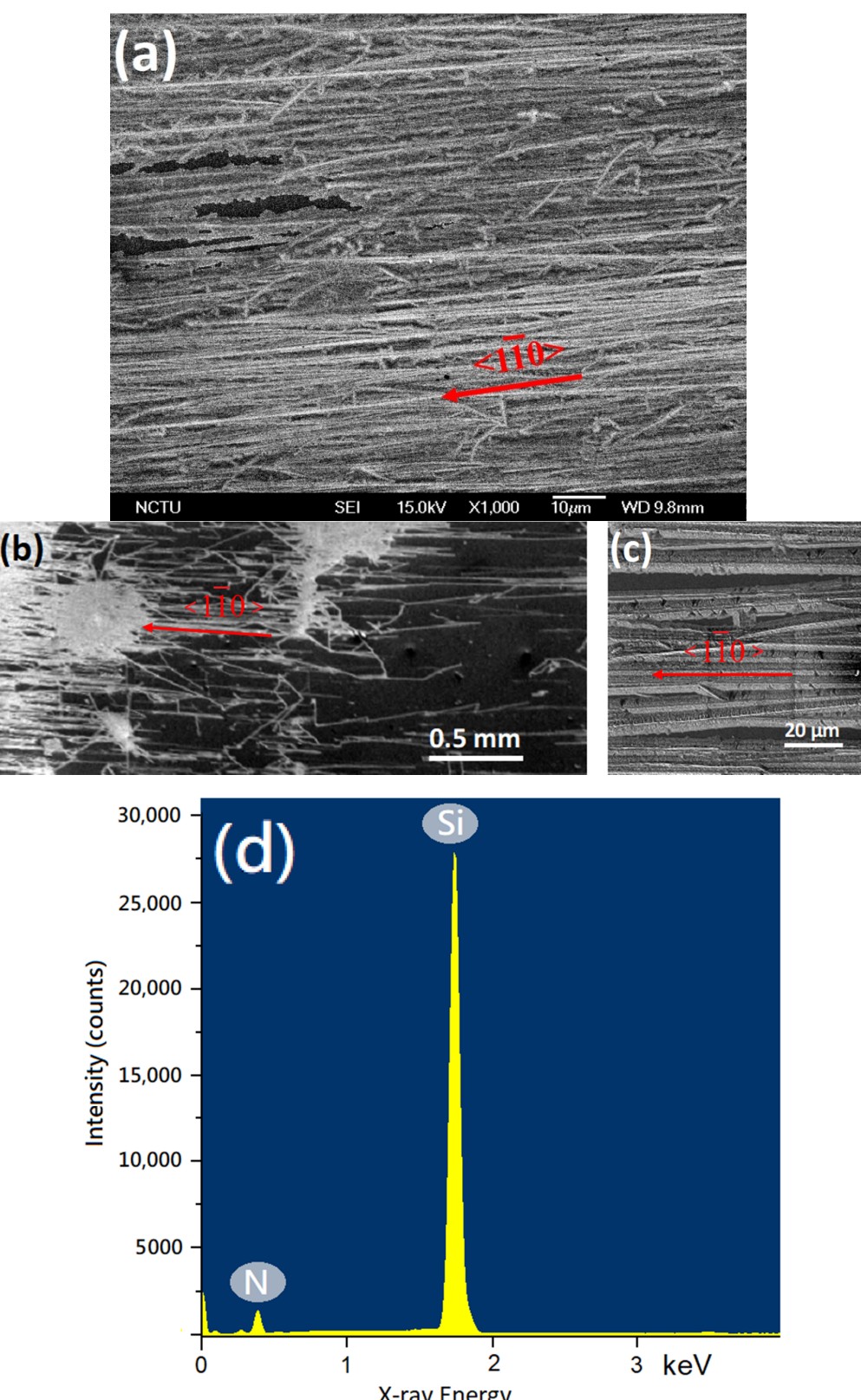

**Figure 1.** SEM surface morphology after nitriding for (**a**) 10 min and (**b**) 20 min. (**c**) Enlarged view of a dense region. (**d**) EDS spectrum of microfibers.

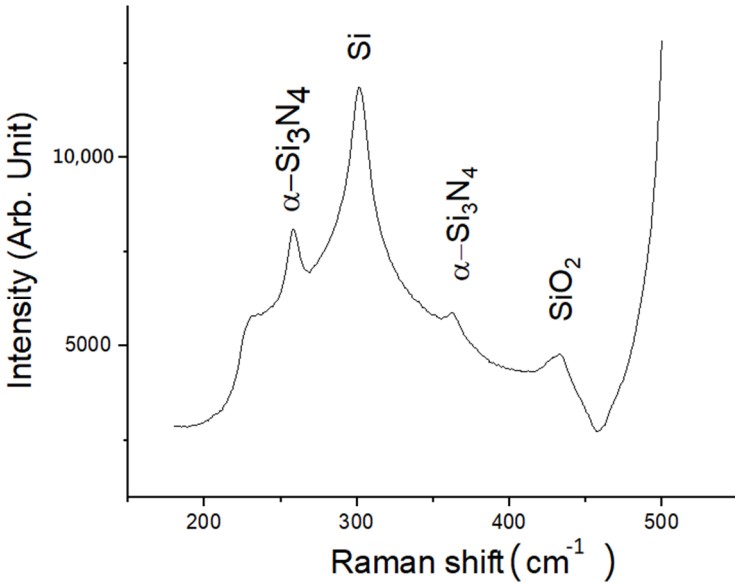

**Figure 2.** Raman spectrum of a nitrided substrate of $SiO_2/Si$ after nitriding for 20 min.

Figure 3 shows typical cross-sectional TEM/STEM images from the nitrided sample for 20 min nitridation after tilting about 7° away from the Si [$1\bar{1}0$] zone axis with the edge-on orientation for the $Si_3N_4$/Si interface. Figure 3a,b show low-magnification annular bright field (ABF) and annular dark field (ADF) images in diffraction and atomic number contrast, respectively. As can be seen, the $Si_3N_4$ thickness is varied from 70 to 270 nm, and no interlayers exist between $Si_3N_4$ and Si. Figure 3c shows a typical high-resolution TEM (HRTEM) image taken from the central region in Figure 3a where the thickness is about 270 nm after tilting about 2° away from the Si [$1\bar{1}0$] zone axis to align $Si_3N_4$ in [$2\bar{1}\bar{1}0$] the zone axis. The HRTEM image shows that $Si_3N_4$ is in direct contact with Si though the interface between them is not flat. The corresponding fast Fourier transform (FFT) pattern from $Si_3N_4$ shown in the upper inset demonstrates that $Si_3N_4$ is an α-phase in [$2\bar{1}\bar{1}0$] the zone axis, whereas the Si FFT pattern shown in the lower inset is deviated from the [$1\bar{1}0$] zone axis. Nevertheless, it can still recognize the pattern characteristics of Si with those reflections to understand the essential orientations with respect to $Si_3N_4$ ones. Furthermore, an atomic resolution ADF STEM image, as shown in Figure 3d, with the corresponding FFT pattern of $Si_3N_4$ in [$2\bar{1}\bar{1}0$] zone axis reveals detailed information at the interface where the $Si_3N_4$ lattice can have a reasonably good match with the Si one, i.e., {$10\bar{1}0$} interplanar spacing of 0.672 nm is correspondingly matched with two Si {111} ones of 0.628 nm, implying that there may exist a specific orientation relationship between $Si_3N_4$ with Si. In addition, it is seen that the zone axis of α-$Si_3N_4$ [$2\bar{1}\bar{1}0$] is approximately parallel to Si [$1\bar{1}0$] with about 2° deviation, suggesting that the lattice mismatch along such a direction is small. The small lattice mismatch can be understood as α-$Si_3N_4$, has lattice parameter *a* = 7.765 Å in [$2\bar{1}\bar{1}0$], and the magnitude of the Si [$1\bar{1}0$] direction is 7.681 Å. Linear fits yield the Young's modulus, *E* = 339 GPa, and the Poisson's ratio, ν = 0.32 [25].

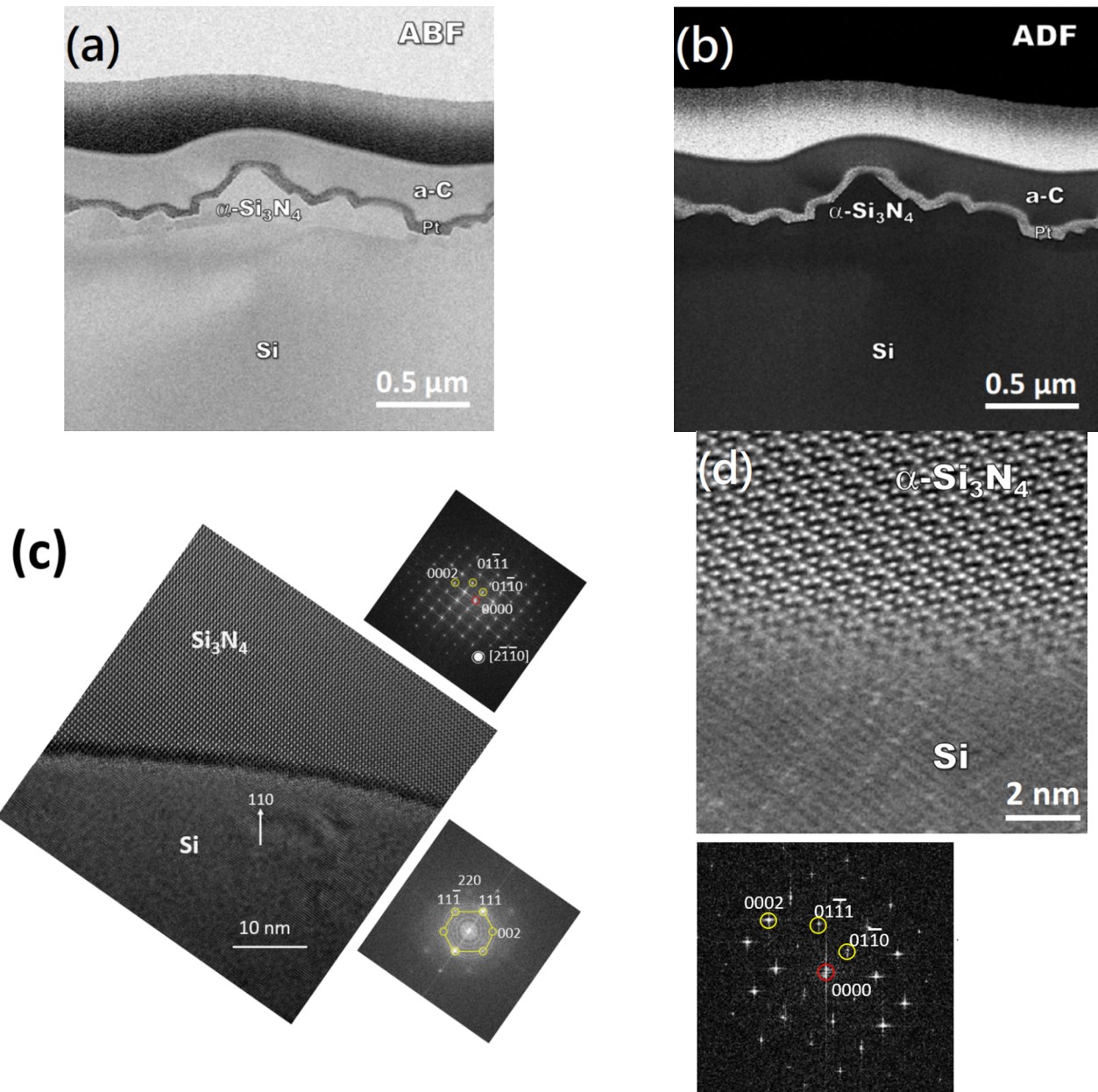

**Figure 3.** (**a**,**b**) Low magnification cross-sectional STEM–ABF and ADF images of $\alpha$-Si$_3$N$_4$ on Si, respectively; (**c**) HRTEM image of the Si$_3$N$_4$/Si interface from the central region in (**a**) with the corresponding FFT patterns of $\alpha$-Si$_3$N$_4$ and Si; (**d**) Atomic resolution STEM-ADF image of the $\alpha$-Si$_3$N$_4$/Si interface with the corresponding FFT pattern of $\alpha$-Si$_3$N$_4$ in zone axis [11$\bar{2}$0].

## 4. Discussion

From the above results, a number of issues on Si$_3$N$_4$ formation will be discussed, including orientation and phase on Si substrate (110) versus (111) the role of SiO$_2$, plasma condition for nitridation, and fiber morphology.

Though Raman spectra and TEM examinations clearly show the formation of crystalline $\alpha$-Si$_3$N$_4$, no Si$_3$N$_4$ diffraction peaks from the samples can be observed in acquired $\theta$-2$\theta$ X-ray diffraction (XRD) patterns in the range of 10–90°, implying that those microfibers are in a large inclined angle away from those diffraction reflections of low indices, such as 10$\bar{1}$0, 10$\bar{1}$1, 11$\bar{2}$0, 20$\bar{2}$1, and 0002. From the FFT patterns of $\alpha$-Si$_3$N$_4$ in Figure 3, 10$\bar{1}$1 reflection is deviated about 7° from the Si surface normal, such that its diffraction signal could not be detected in general $\theta$-2$\theta$ XRD scan.

It is known that α-to-β transformation takes place at high temperatures (~1900 °C) and the process is irreversible. It has been shown that β-$Si_3N_4$ is thermodynamically more stable than α-$Si_3N_4$ up to 2000 K [26], and α-$Si_3N_4$ is a metastable phase under ordinary pressure. However, plasma nitridation can be considered as a nonequilibrium condition to form α-$Si_3N_4$. Therefore, it might be the dominant phase for the short time of nitridation. Currently, it is not known why the formation of only α-$Si_3N_4$ occurs on Si (110), while both α- and β phases are formed on Si (111), even if the surface energy of Si (110) is higher than Si (111), which might lead to a lower energy barrier for α-$Si_3N_4$ nucleation [27]. However, it is worth pointing out that most of previous studies on nitridation for formation of $Si_3N_4$ nanowires and fibers show only an α-phase formed [15,28].

Direct nitridation of bare Si substrate without $SiO_2$ under the same plasma condition for 20 min shows no nitride formation, as no Raman signals can be observed and SEM observations show that the surface of the Si substrate remains to be as smooth as that before nitridation. $Si_3N_4$ formation is slow for nitridation of Si and the thickness formed is only a few nanometers after long-time nitriding, while the nitridation rate is faster with $SiO_2$ to form a thicker nitride film [29]. Thus, it is likely that the reaction of Si with N from the plasma will be retarded with the presence of H in plasma, and it might require an extended period of plasma nitridation to obtain $Si_3N_4$.

For thermal nitridation, Si and $SiO_2$ can react to form SiO gas species which then react with nitrogen for nitride formation as the following reactions of (1) and (2) [30].

$$Si_{(s)} + SiO_{2(s)} \rightarrow 2SiO_{(g)} \tag{1}$$

$$6SiO_{(g)} + 8N_{2(g)} \rightarrow 2Si_3N_{4(s)} + 3O_{2(g)} \tag{2}$$

It has been reported that silicon oxide can be nitrided easily below 700 °C using nitrogen plasma generated by electron impact and a large amount of nitrogen can be incorporated in the oxide [5,31]. As most of the gas phases for nitridation can be effectively decomposed in the microwave plasma, there may be plenty of atoms, radicals, and ions as reactive species which can be available for the reactions on the substrate surface. The chemical reactions might occur with a faster rate under the microwave plasma condition. In addition, microwave plasma with H can effectively reduce $SiO_2$ to Si which then reacts with N. In particular, atomic hydrogen may play a critical role on nitridation. The diffusion rate of atomic hydrogen from the plasma might be faster than that of nitrogen toward the $SiO_2$/Si interface to delaminate the oxide layer, whereas hydrogen can reduce the oxide surface to expose the bare Si surface which may have a slow reaction with N in comparison with the SiO reaction with N to form $Si_3N_4$ [32,33]. The accelerated nitriding rates due to the interaction of hydrogen with native oxide on the surface of the Si particles were observed in previous studies [34,35]. Similarly, $SiO_2$ can be transformed to $Si_3N_4$ by $N_2/H_2$ microwave plasma in the present case. In addition, it can be seen that pure $N_2$ plasma nitridation of $SiO_2$ may result in the formation of SiON instead of $Si_3N_4$.

Nitridation of Si (110) has been previously investigated. Saranin et al. reported that the Si(110) surface after thermal nitridation by $NH_3$ gas in the temperature range of 560–1050 °C is covered with epitaxial islands of Si-nitride in a thickness less than 1 nm, followed by layer-by-layer growth [36]. Higuchi et al. also reported that high-quality $Si_3N_4$ film in a thickness < 5 nm was formed by nitridation of the Si(110) surface at 600 °C using radical NH [5]. Also, nitridation of Si(110) under RF plasma and the thermal process forms a very thin ($10\bar{1}1$) α-$Si_3N_4$ epitaxial layer [37]. In contrast, the present work shows that on Si (110), the α-$Si_3N_4$ microfibers only align along with the single direction of Si [$1\bar{1}0$] after nitridation by microwave plasma. Though the exact mechanism for α-$Si_3N_4$ formation is not currently known, it may be of interest to understand the relationship from the crystallography point of view. While $SiO_2$ can be decomposed by $N_2/H_2$ plasma, the nucleation and growth of $Si_3N_4$ could be fast along Si <110> due to low strain energy from the small lattice mismatch between them along Si <110> and $Si_3N_4$ <11$\bar{2}$0>.

Nitridation of Si and $SiO_2$ to form nanowires and nanofibers were previously reported. Ramesh and Rao reported formation of $\alpha$-$Si_3N_4$ fibers by carbothermal reduction and nitridation reaction of $SiO_2$ with $N_2$ gas at 1623K [38]. Also, a few reports have shown synthesis of $\alpha$-$Si_3N_4$ nanowires by nitriding nanocrystalline Si powder at 1300 °C for 2 h in pure $N_2$ [39]. Kim et al. demonstrated synthesis of silicon nitride nanowires directly from the silicon substrates via a catalytic reaction under ammonia or hydrogen flow at 1200 °C, using Ga, GaN, and Fe nanoparticles as catalysts [29]. However, those nanowires are randomly distributed. Formation of $\alpha$-$Si_3N_4$ whiskers by nitridation have been also reported by previous studies [35,40,41]. The synthesis of $Si_3N_4$ nanowires from the reaction of Si nanoparticles with $N_2$ in the temperature range of 1200–1440 °C is also reported [42–44]. Ordered arrays of $\alpha$-$Si_3N_4$ nanowires have been also synthesized at 1150 °C with $NH_3$ [45,46]. Direct synthesis of $\alpha$- $Si_3N_4$ nanowires from silicon monoxide on alumina with $N_2/H_2$ was also reported [47]. Similarly, it has been demonstrated that single-crystalline $\alpha$-$Si_3N_4$ nanowires can grow in a direction perpendicular to the wet-etched trenches in the SiO film on the plane of the Si substrate without metal catalysis [33].

## 5. Conclusions

Microwave plasma nitriding of 50 nm $SiO_2$ on Si(110) substrate using a gas mixture of $H_2$ and $N_2$ can form aligned microfibers of crystalline $\alpha$-$Si_3N_4$ nearly along with Si [1$\bar{1}$0]. The formed $Si_3N_4$ microfibers can have a length as long as 2 mm with a diameter of 5–10 μm. Cross-sectional STEM examinations reveal that $Si_3N_4$ directly forms on Si without any residual $SiO_2$ after nitriding. Furthermore, it is shown that most of the microfibers have their longitudinal direction along $Si_3N_4$ <11$\bar{2}$0>, which is approximately parallel to Si [1$\bar{1}$0].

**Author Contributions:** Conceptualization, C.-H.Y. and L.C.; Data curation, C.-H.Y., K.-A.C. and W.-C.C.; Formal analysis, C.-H.Y., T.-H.D. and K.-A.C.; Funding Acquisition, W.-C.C. and L.C.; Methodology, C.-H.Y.; Software, T.-H.D.; Supervision, L.C.; Validation, C.-H.Y.; Writing—original draft, C.-H.Y.; Writing—review & editing, L.C. All authors have read and agreed to the published version of the manuscript.

**Funding:** This work was supported by Ministry of Science and Technology, Taiwan, R.O.C. under contract of MOST 107-2221-E-009-009-MY2.

**Institutional Review Board Statement:** Not applicable.

**Informed Consent Statement:** Not applicable.

**Data Availability Statement:** Data sharing is not applicable to this article.

**Conflicts of Interest:** The authors declare no conflict of interest.

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
