# Peer review of "Formation of Aligned α-Si3N4 Microfibers by Plasma Nitridation of Si (110) Substrate Coated with SiO2"

_coatings, doi:10.3390/coatings11101251_

Round 1

Reviewer 1 Report

Opinion on the paper: “ Formation of aligned α-Si3N4 microfibers by plasma nitridation of SiO2 on Si (110) substrate”

The paper is on α-Si3N4 microfibers by plasma nitridation of SiO2 on Si (110) substrate. The subject is interesting, however the subject of the work is very similar to the one recently published in the article by the same authors: (C.H. Yu, K.A. Chiu, T.H. Do, L. Chang, Oriented Si3N4 crystallites formed by plasma nitriding of SiO2/Si (111) substrate, Surf. Coat. Technol. 395 (2020) 125877. https://doi.org/10.1016/j.surfcoat.2020.125877). The introductions are almost identical in the both papers. I think, it is a kind of self-plagiarism.

The paper can be published but changes must be made. Introduction cannot be the same as in the earlier work. It should be completely changed to introduce new information. The authors should advise why they decided to conduct similar studies to those described earlier. The publication submitted for review and the article published earlier differ only in the type of silicon substrate: Si (110) in this paper and Si (111) in the work published earlier. Do the authors believe that the method of obtaining a Si3N4 presented by them is better than other numerous methods previously described in the literature? If so, they should show it in the article.

Author Response

We thank the reviewer for the valuable comments and suggestions. We have modified some of the content. There is a differences between Si(111) and Si(110). For Si(111) on the Si3N4 on the surface is mainly and arranged at an angle of about 60º. For Si(110),  the Si3N4 on the surface is approximately parallel to Si < 1-10>. Therefore, the ratio of the α-Si3N4, β-Si3N4 is different.

Reviewer 2 Report

Please see the comments in the document attached. 

Author Response

We thank the reviewer for the valuable comments and suggestions.
The PDF file answers the questions. 
Please see the attachment

Reviewer 3 Report

Dear authors, dear editor,

I read the manuscript "Formation of aligned \alpha-Si_3N_4 microfibers by plasma nitridation of SiO_2 on Si (110) substrate. It is reported an investigation of the structure of the microfibers by means of STEM and Raman spectroscopy. I did not find any serious flaws within this analysis. In the end I find it is a pity, that the authors did not really try to do experiments in order to understand the formation mechanism of the microfibers, though there is a discussion about it.
E.g. one could have changed the plasma Ar instead of H, possibly only H (how is the structure look like afterwards). Such experiments could elucidate a bit more the formation mechanism.
Therefore I think that one should only accept the paper after such additions in order to get more substance into the paper.

Some minor issues:
- in the abstract temperature -> temperatures
- due to the structure of Si3N4, there should be 2 different lattice strains , please give values.

Author Response

We thank the reviewer for the valuable comments and suggestions.

We completely agree with the point on "understand the formation mechanism of the microfibers". However, it may take a great effort to obtain solid evidence from experimental data, and a quite number of plasma parameters for nitriding need to try such as the reviewer's suggestion of using Ar. Also, the temperature and pretreatments of Si surface, different coating of SiON on Si, etc., are taken in consideration. As no one has done microwave plasma nitriding on Si for which our current understanding may be in the very early stage, the manuscript presents what we have observed so far.

Round 2

Reviewer 1 Report

The paper should be accepted in the present form.

Author Response

We are grateful to the reviewer for the valuable comments.

Reviewer 2 Report

Small error, please change. in the cover letter, you agree that it should be diameter instead of width. but this is not changed in the manuscript. Please change. 

Author Response

We apologize for "width" caused by our mistake.

We are grateful to the reviewer for the valuable comments.

Reviewer 3 Report

Already in my first report I had not so much to crititicize. Only that I would have wished further experiments to illucidate more the  microfiber formation mechanism. But of course, I agree, that this might be not so easy and fast done. In this respect I recommend the paper for publication in the present form.

Author Response

(The authors gave the same response as above.)
